# Expressing uncertainty in Human-Robot interaction

**Christoph Bartneck** **\*, Elena Moltchanova**

University of Canterbury, Christchurch, New Zealand

☺ These authors contributed equally to this work.
\* christoph.bartneck@canterbury.ac.nz

**Data Availability Statement:** All relevant data are available within the Open Science Foundation at URL: https://osf.io/7fz6v/ and DOI: 10.17605/OSF.IO/7FZ6V.

**Funding:** We received no funding for this study.

## Abstract

Most people struggle to understand probability which is an issue for Human-Robot Interaction (HRI) researchers who need to communicate risks and uncertainties to the participants in their studies, the media and policy makers. Previous work showed that even the use of numerical values to express probabilities does not guarantee an accurate understanding by laypeople. We therefore investigate if words can be used to communicate probability, such as "likely" and "almost certainly not". We embedded these phrases in the context of the usage of autonomous vehicles. The results show that the association of phrases to percentages is not random and there is a preferred order of phrases. The association is, however, not as consistent as hoped for. Hence, it would be advisable to complement the use of words with numerical expression of uncertainty. This study provides an empirically verified list of probabilities phrases that HRI researchers can use to complement the numerical values.

## Introduction

When HRI researchers apply for funding, talk to the media, or ask participants for their consent, they are forced to communicate with an audience that might not be trained or knowledgeable in statistics. It is therefore inherently difficult to communicate uncertainty and the associated risks to these audiences. Previous worked showed that numerical expression of uncertainty is not optimal for communicating with children [1] and that even older adults consider pictorial representations to be a good alternative [2].

Reporters may ask about the application and usefulness of HRI research on a societal level. At times they may even confront HRI researchers with predictions from popular science fiction movies. Responding with statistically accurate answers to such questions is difficult. This is not only because the researcher might not have the answers to such high-level questions, but also because the audience might not be able to understand the statistics. Moreover, the general public might even have an underlying fear of robots that might influence their perception [3].

The possibly most important communication challenge for HRI researchers is talking to politicians who intend to legislate the use and application of robots. Highly controversial topics, such as the personhood of artificial intelligence [4], the use of sex robots [5] or

**Competing interests:** The authors have declared that no competing interests exist.

autonomous vehicles [6], are currently being discussed despite the absence of sufficient empirical evidence.

In all of these instances HRI researchers need to communicate with readers who are not trained in scientific methods or statistics. In an interview with a reporter it makes little sense to quote p-values or confidence intervals. [7] pointed out that "numbers do not speak for themselves; the context, language and graphic design all contribute to the way the communication is received." Politicians are likely to ask for simple answers about the impact that robots might have on society. They want to be the person with just one clock who knows the exact time rather than a person with two who is forced to be uncertain.

A dramatic example of the importance of appropriate communication to non-experts is the 2009 L'Aquila earthquake that killed several hundred Italians. Seven seismologists and geologists, who were members of the National Commission for the Forecast and Prevention of Major Risks, met six days before the quake struck. Despite a series of small tremors they did *not* issue a safety warning. In subsequent prosecutions that lasted for up to seven years they were initially sentenced in court for involuntary manslaughter. In the end, however, all the researchers and government officials were acquitted.

This court case sent shock waves through the scientific community. If scientists would have to face legal consequences for their predictions and viewpoints then hardly anybody would dare to speak out, let alone in language that refrains from the heavy use of statistics and highly restrained conclusions. HRI researchers, too, could be held responsible for not warning society about the foreseeable and unforeseeable effects of robots, such as the loss of jobs and the loss of lives due to accidents with autonomous vehicles.

There is, of course, nothing wrong with using statistics to express the results of a study, but they remain largely inaccessible for the general public. Refraining from trying to communicate with the general public is also not an option since it would likely result in the ivory tower syndrome in which science operates far removed from societal problems. Wittgenstein's suggestion that "Whereof one cannot speak, thereof one must be silent" [8] also does not seem to apply since although it might be difficult to understand the concepts of probability, it is not impossible and it is based on facts.

Communicating uncertainty and risk is one of the most challenging science communication tasks. Not only are people particularly bad at making decisions based on probabilities [9], but the perception of risk itself is subject to a variety of biases [10]. A 5% mortality chance is perceived very differently from a 95% survival chance [7]. Moreover, the actual probabilities, such as the ones associated with causes of death, are very different to what people search on Google, which are again very different to any resulting media coverage (see Fig 1). While there is only a < 0.01% chance of dying due to a terrorist attack, 7.2% of Americans search this on Google and terrorism receives around 34% of media coverage.

Previous work highlighted some of the challenges in risk communication but remained on the theoretical level [11]; [12]. [13] provided a review of studies that analysed probabilistic expressions. In the carefully drafted literature section the authors included information about the sample populations. It becomes clear that almost all studies reviewed used participants that were either domain experts (e.g. judges, medical doctors), trained in statistics or both. The same holds true for several other studies [14]; [15]. For this study we therefore used Amazon Mechanical Turk (MTurk or AMT) to sample the general population. [16] presented demographics of the AMT population and concluded that AMT workers are more representative of the general population and substantially less expensive to recruit.

This discrepancy between actual probabilities, perceived risks and media coverage can have negative effects not only on the development of certain technologies, but also on the actual loss of lives. Arguably, autonomous vehicles have a far better driving record than human drivers

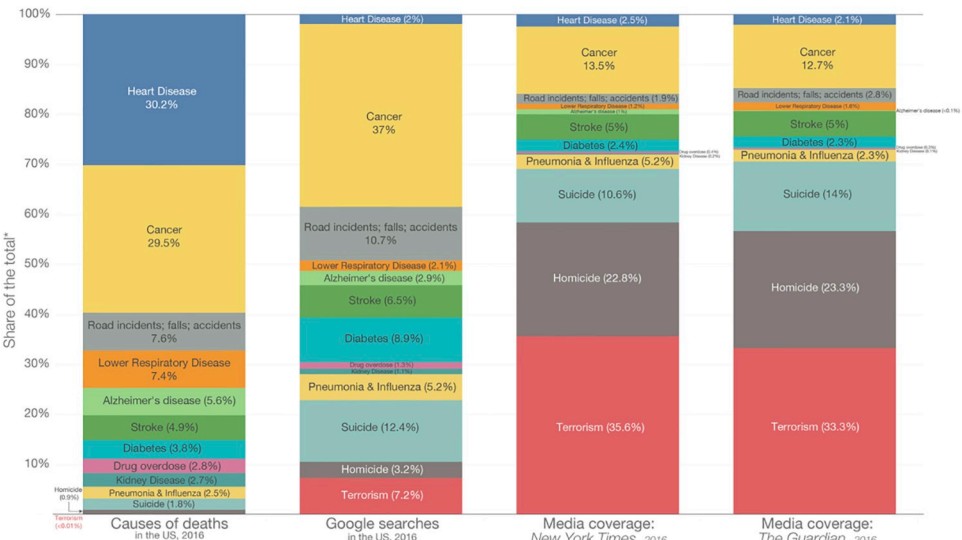

**Fig 1. What Americans die from, what they search on Google and what the media reports on.** (source https://ourworldindata.org/does-the-news-reflect-what-we-die-from). Permission under the Creative Commons CC-BY license.

[17], although the exact definition of an accurate measurement for the safety of autonomous vehicles remains disputed [18]. Robert Sparrow and Mark Howard even proposed that it will become illegal for humans to drive cars once autonomous vehicles outperform humans [6].

The fatal accident in which an autonomous Uber car killed the pedestrian Elaine Herzberg received enormous media attention. Elon Musk already argued that such disproportionate media coverage could inhibit the adoption of autonomous vehicles which in turn would lead to many unnecessary fatal accidents caused by human drivers [19].

It is important to distinguish between risk and uncertainty in public communication. While risk focuses on negative events, uncertainty also includes positive outcomes. In our study we therefore decided to focus on uncertainty since autonomous vehicles can produce positive and negative outcomes. We investigate how uncertainty can be communicated to the general public without the use of scientific statistics. More specifically, we investigated how uncertainty can be expressed in words such as "likely" and "almost certainly not". It is necessary to understand how these phrases are being translated to numerical expression of uncertainty and how well this matches the reverse operation in which numerical expression of uncertainty is being translated back to phrases. These uncertainty phrases can be embedded in the context of autonomous vehicles, which are currently the most widely adopted human-robot interaction systems targeted at the general public. The context to which a probability refers does play a role in the communication [20]; [21].

An example sentence could be "There is almost no chance that your autonomous vehicle avoids the collision with another car." Here we encounter a methodological problem. The sentence above expresses a very high certainty and at the same time a very low probability of the event (avoiding the collision) to occur. The sentence declares with high certainty that the event is unlikely to happen.

Non-experts might fall into the trap of using the terms probability, likelihood and certainty as synonyms. Whereas statistically, given the probability distribution of an event, the probability refers to the parameter of that distribution, the likelihood is a function of the parameter given the observed data, and uncertainty is associated with accuracy of either measurement,

estimation or prediction. It is therefore necessary to investigate if laypeople use the words certainty and likelihood interchangeably. The research questions therefore are:

1. How consistently do words express uncertainty and likelihood?

2. Does the context provided in a sentence influence its perception of uncertainty and likelihood?

 a. What influence does the desirability of the consequences have on the perception of its uncertainty and likelihood?

 b. What influence does the agent of the action have on the perception of its uncertainty and likelihood?

## Method

We conducted an experiment in which the phrases used to express uncertainty (phrase), the direction of the translation (direction), and the desirability of the consequences of an event (valence) are the within participants factors. The agent responsible for the event (agent) and the association word (prompt) are the between participants factors. The two prompts were "uncertainty" and "likelihood".

We used 16 different phrases that were grouped into four categories (very certain, certain, uncertain, very uncertain). We used 16 different consequences that were grouped into four categories (very desirable, desirable, undesirable, very undesirable). We used two directions of translations. Phrases were assigned to percentages or percentages were assigned to phrases. The percentages were either percentages of certainty or likelihood. The agent of the action was either the human ("you") or an autonomous vehicle ("your autonomous vehicle").

In addition, we asked the participants to associate the phrases to uncertainties and likelihood in the absence of any context. We also asked the participants to order the context-free phrases according to their uncertainty and likelihood.

Due to the possibly complicated relationship between the variables involved and aiming for ease of interpretation, we decided to forego the conventional categorical and ordinal multinomial logistic regression techniques and used the non-parametric classification and regression trees (CART) instead. The interested reader might consult [22] for further details on this statistical approach. Building a tree entails finding a set of if-then decision making rules. Unlike regression models, there is no assumption of linearity in parameters or even monotonic relationship between the independent variables and the expected response. This is especially useful in this study, because we do not have a priori knowledge of the likely structure of effects of context and valence on the probability interpretation. In addition, an ordinal multinomial logistic regression usually requires the assumption of proportional hazards [23], which often does not hold. Finally, multinomial logistic regression often suffers from separation, which leads to convergence issues [24]. Separation is especially likely in case of multinomial response and a large number of categorical predictors, all with many categories, as is the case in this study.

To summarize, the trees do not require any distributional assumptions, do not suffer from convergence issues, and the resulting model can be represented in the form of an easy-to-understand flow chart or a rules table. Ensembles of trees constitute random forests, which tend to work better than individual trees, but become harder to interpret. [25]. To avoid overfitting, the models are judged on the basis of goodness-of-prediction rather than goodness-of-fit. The former is evaluated based on cross-validation [22], where the *training* part of the data

is used to fit the model, which is then used to predict the response of interest for the *testing* part, and the measure of goodness-of-prediction is produced based on this result.

For the categorical responses, it is common to use confusion matrices (tables of observed vs. predicted responses) and apparent error rates (the proportion of incorrect predictions) to evaluate model performance.

Trees and Random forests belong to the data mining techniques, where the practitioners are not particularly worried about statistical significance. The complexity of the model, which is determined by the number of splits for the tree and the size of the ensemble for the random forest, is tuned using cross-validation, and the importance of explanatory variables is assessed based on whether or not they are included in the final model. The importance score is evaluated for each variable to show what effect it has on the predictive power of the model. In situations where predictor variables are correlated or include factors with a widely different number of categories, an alternative, conditional importance index can be computed [26]. While there is no method to obtain *p*-values for the effect of variables included in trees, it is possible to obtain permutation test-based *p*-values for the random forest [27]. All the analyses were implemented in R, using packages rpart [28], rparty [26], randomForest [29], and rfUtilities [30].

This study has been approved by the Human Ethics Committee of the University of Canterbury under the reference HEC 2019/30/LR-PS.

## Participants

We recruited 297 participants from Amazon Mechanical Turk (MTurk). Participants received 1 USD for their participation which took approximately 10 minutes. A total of 297 people (171 men and 126 women) participated in the study. The age of participants ranged from 22 to 72 years of age for men ($M = 37.4$, $Md = 35.0$, $SD = 8.8$) and from 23 to 69 years of age for women ($M = 43.4$, $Md = 39.5$, $SD = 11.7$). All participants were native English speakers and lived in the USA, UK, Canada, Ireland, Australia or New Zealand. All participants were Amazon Mechanical Turk Master Workers. These workers are being monitored by Amazon for their performance over time. Amazon explains that "Workers who have demonstrated excellence across a wide range of tasks are awarded the Masters Qualification. Masters must continue to pass our statistical monitoring to retain their qualification".

Previous studies have indicated that data collected via MTurk is of equal quality as on-campus recruitment or participant data from forums [31]; [32]. The MTurk population is more representative of the general public than (under)graduate samples and they produce reliable responses at an affordable price [33]. Using MTurk to recruit participants has established itself as a valid methodology and the number of studies that use this method grows quickly [33]. While there are undoubtedly many other factors that would deserve further research, out study had to focus on a manageable number of questions. We therefore need to point the interested reader to other studies that specifically investigated the characteristics of MTurk workers. Previous studies investigated the demographic characteristics, political preferences, occupation and race [34]. [35] studied MTurkers' personality and psychopathology features and concluded that they provide sufficiently high quality responses and that they are reasonably representative. A recent study highlighted the potential decrease in the quality of responses solicited through MTurk but pointed out that this can be mitigated though data validation and attention checks [36].

## Process

After completing a short list of demographic questions, the participants were randomly assigned to either the context-free first sequence or the context-included first sequence. They

were then randomly assigned to the human or the autonomous vehicle conditions. They were then randomly assigned to either translating phrases to percentages first or the reverse. Participants then randomly received one out of each of the four probability categories which each consisted of four phrases. The participants in the human condition might, for example, receive the phrases:

- It is almost certain that you. . .

- It is probably that you. . .

- It is improbable that you. . .

- There is almost no chance that you. . .

These phrases were then randomly combined with one out of each of the four valence consequences. Here is an example for one combination:

- It is almost certain that you avoid the collision with another car.

- It is almost certain that you use the fastest route to the destination.

- It is almost certain that you use a lot of fuel for the drive.

- It is almost certain that you harm other traffic members

The participants thereby translated $4 \times 4 = 16$ sentences to percentages. In the reverse condition, percentages to sentences, the participants were given a percentage and were asked to associate a sentence to it, such as:

What sentence below expresses 20% certainty?

1. It is almost certain that you drive safely.

2. It is highly likely that you drive safely.

3. There is a very good chance that you drive safely.

4. It is probable that you drive safely.

5. It is likely that you drive safely.

6. You are probably going to drive safely.

7. We believe that you drive safely.

8. The chances are better than even that you drive safely.

9. We doubt that you drive safely.

10. It is improbable that you drive safely.

11. It is unlikely that you drive safely.

12. You are probably not going to drive safely.

13. There is little chance that you drive safely.

14. There is almost no chance you drive safely.

15. It is highly unlikely that you drive safely.

16. Chances are slight that you drive safely.

The list above is ordered by the sequence provided by [37]. Their numbering will be used in the results section to specifically talk about a phrase.

Each participants had to associate all of the percentages (0%-100% in 10% steps) in a randomised order to sentences. The sentences were randomly selected from the 16 possible. The participants had to answer 16 + 11 = 27 questions. In addition, we asked three demographic and two context-free questions, which brings the total to 32 questions.

For the context-free associations of the phrases to certainties, the order of the phrases was randomised. We then asked the participants in a second step to sort the phrases according to their certainty. The starting order of the phrases was randomised.

In a second data collecting session, we repeated all the steps above but instead of using the "uncertainty" prompt we used the "likelihood" prompt. We used Qualtrics Ballot Box Stuffing prevention function to deny participants from the first session access to the second session.

## Stimuli

The 16 phrases to express probability were taken from [37]. The phrases had been translated to percentages by experts in the security domain. We thereby could group the phrases into four categories based on these initial ratings.

The 16 valence ratings were divided into four quartiles. The most positive focus on saving lives and avoiding harm. The second quartile focuses on efficiency, such as using little fuel or finding the fastest route. The two negative quartiles are the opposites of the two positives. They focus on using a lot of fuel and causing harm.

## Measurements

For each participant the resulting data set contains information about his/her age and gender. In addition, the data sets contains data for each direction (sentence-to-percentage and percentage-to-sentence), for one prompt ("likelihood" or "uncertainty"), for one agent ("you" vs. "autonomous car"), for four out of 16 phrases, and for four out of 16 valences. In addition, we have recorded the preferred ordering of the context-free phrases from the likeliest to the least likely.

## Pilot study

We conducted a pilot study with 20 students from a data visualisation course at the University of Canterbury. The students were all trained in statistics and programming. Their nationalities were predominately Chinese and Indian. The students thereby do not reflect the general public, but their contribution was sufficient to test our research methodology and process. The results of this pilot were very scattered which made us concerned about whether their English language skills had been sufficient to fully understand the questions. For the main experiment we therefore limited the participants to native speakers of English.

## Results

In a first step we inspected the completion time of the participants to eliminate any potential fraudulent responses. It took between 90 and 3470 seconds for the participants to finish the survey. No suspiciously low values or outliers were detected. However, the responses of a person with the extremely long overall response time of 117 hours were omitted from further analysis.

The scatter plots of phrases vs. the associated percentages and vice versa for both, the "likelihood" and the "uncertainty" prompt are shown in Figs 2, 3, 4 and 5.

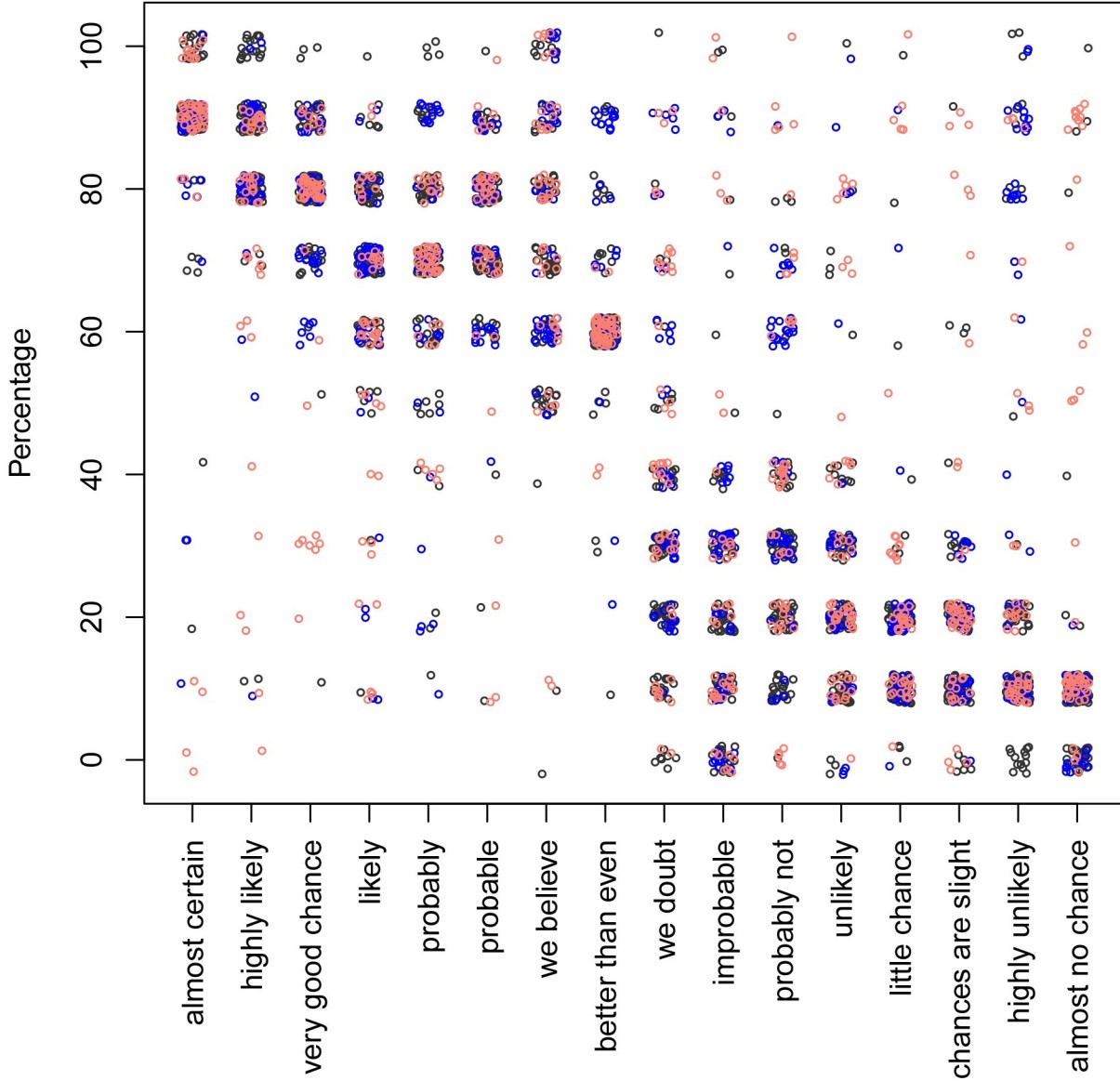

**Fig 2. The percentages associated with the phrases, using the "likelihood" prompt.** The black, blue, and red colours correspond to the context-free, user and car contexts, respectively. Each dot represents one response.

The ordering of the phrases from most probable (1) to least probable (16) as perceived by the survey participants are shown in Fig 6. The intensity of the grey level corresponds to the percentage of respondents who associated each of the phrases with the particular rank. The phrases have been ordered according to the popular preference. It can be seen that the phrases "almost certainly" and "almost no chance" had very firm positions at the opposite ends of the spectrum, and the phrase "better than even" tended to be placed $9^{th}$ but for the rest of the categories the order preference was less clear.

In order to model the dependency of the responses, we have considered three models in each case: (i) the naive model, where the most popular overall response category of phrase/percentage was chosen in response to the percentage/phrase prompt and no other variables were taken into account; (ii) a single classification/regression tree; and (iii) a random forest. We

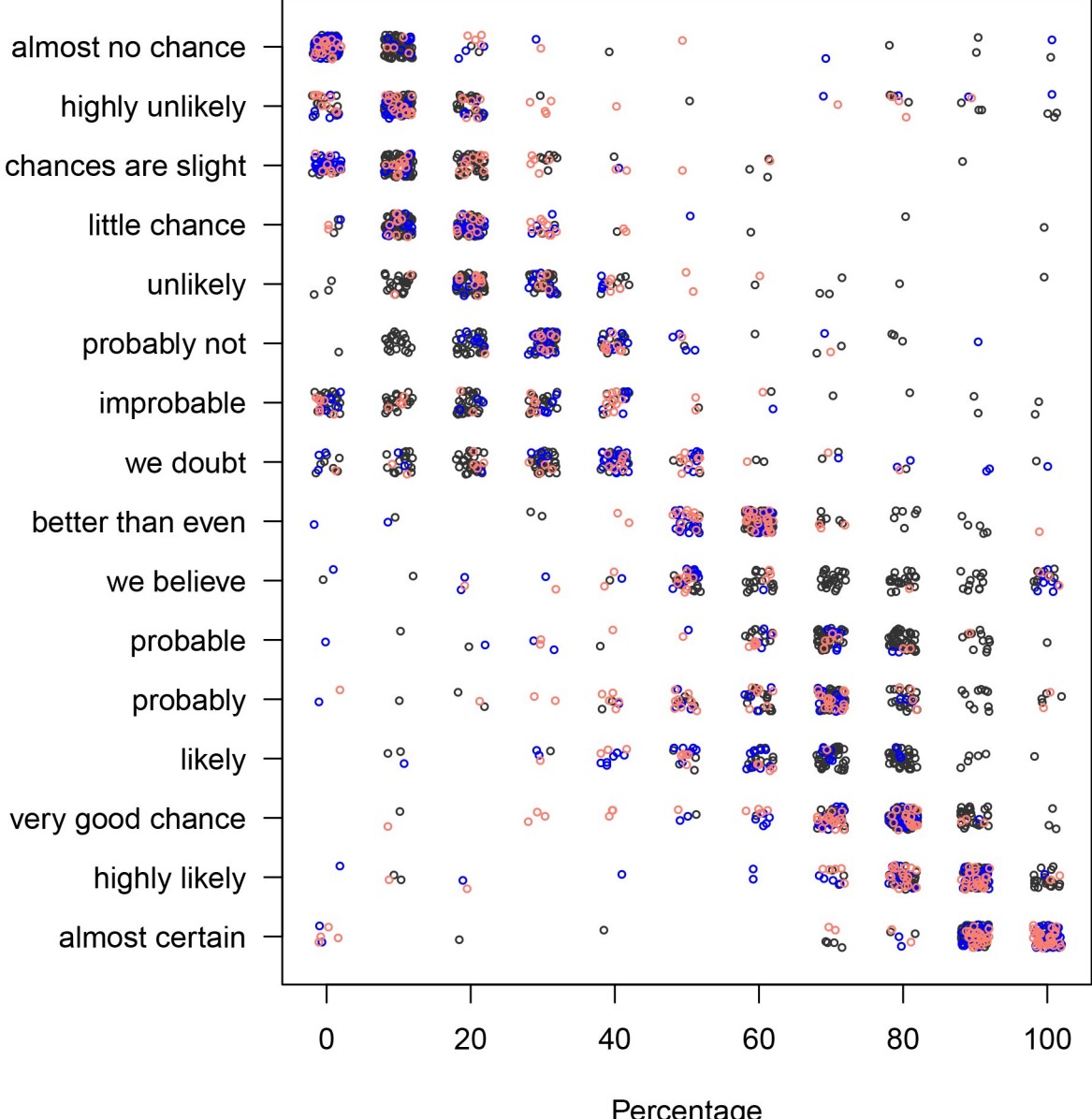

**Fig 3. The phrases associated with the percentages, using the "likelihood" prompt.** The black, blue, and red colours correspond to the context-free, user and car contexts, respectively. Each dot represents one response.

then used 10-out cross-validation to produce a confusion matrix and produce the apparent error rate to within 1, 2, etc categories. The results for sentence-to-percentage with the "likelihood" prompt are shown in Fig 7. There was not much difference between a single tree and a naive model, implying that no other variables than phrase made a difference in prediction accuracy. The random forest was marginally better. When the percentage was treated as a numerical variable, the resulting RMSE was 21.5 and 21.7 for uncertainty and likelihood prompts, respectively.

The apparent error rates (AER) for percentage-to-sentence with the "likelihood" prompt are shown in Fig 8. Here, the random forest, the classification tree, and the naive approach did

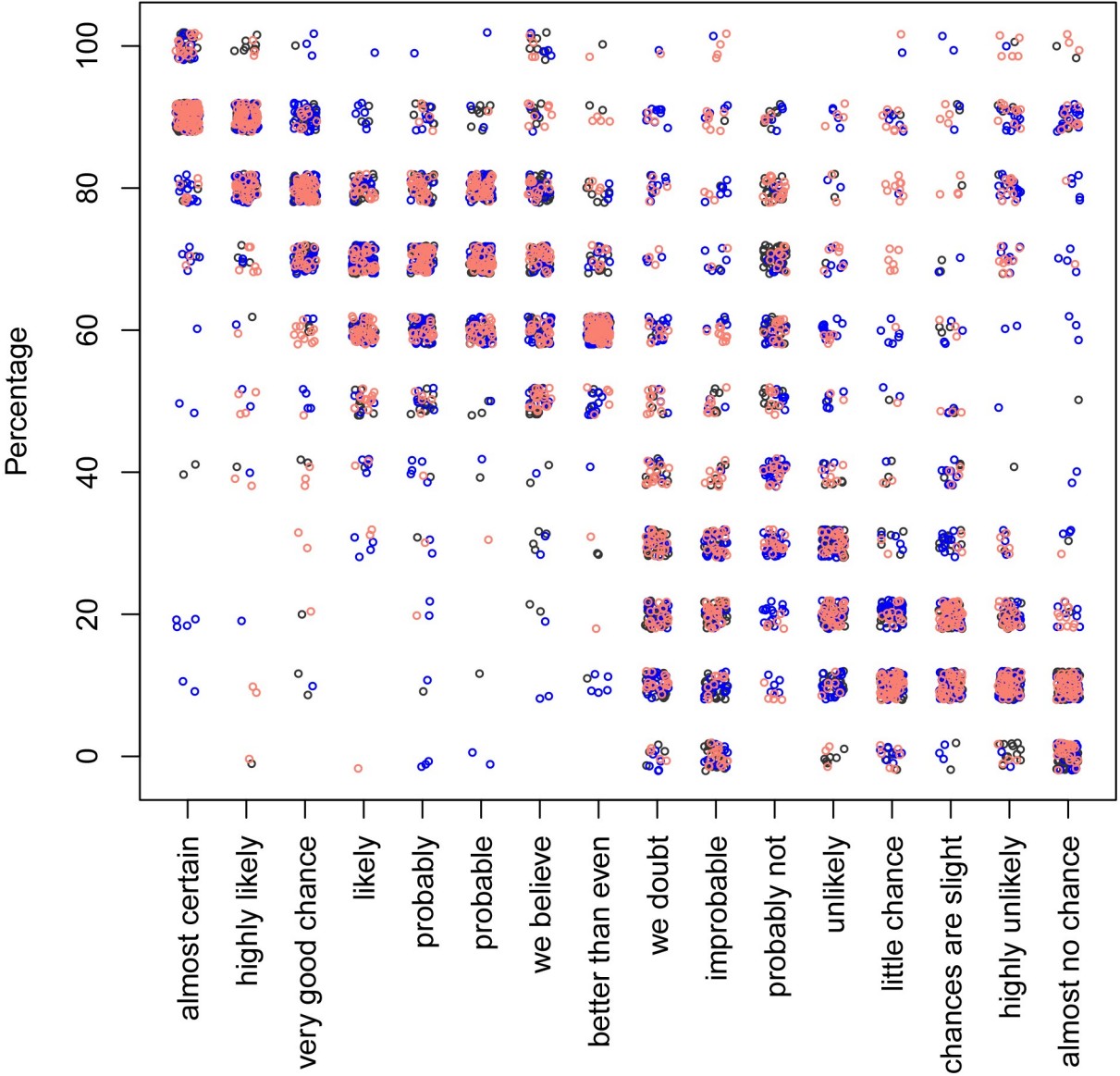

**Fig 4. The percentages associated with the phrases, using the "uncertainty" prompt.** The black, blue, and red colours correspond to the context-free, user and car contexts, respectively. Each dot represents one response.

not differ in terms of prediction accuracy. Although the AER of 75% for predicting the phrase exactly may seem high, it is still substantially lower than the result of random guessing ($\frac{15}{16} \times 100\%$). Furthermore, the probability of getting the category correctly to within, for example, 2 categories as arranged in Fig 6 is quite high at 75%.

In order to find out whether sex, age, and context (i.e. agent and valence) had an effect on the perception of correlation between percentages and phrases, we have studied the importance of the above variables in the random forests, as well as whether or not they were included in the single best tree. Table 1 shows the importance scores of variables in random forest conditional on the variables already added. Note, that as explained in the methods, the importance scores are not p-values and they do not reflect statistical significance of the variable-specific

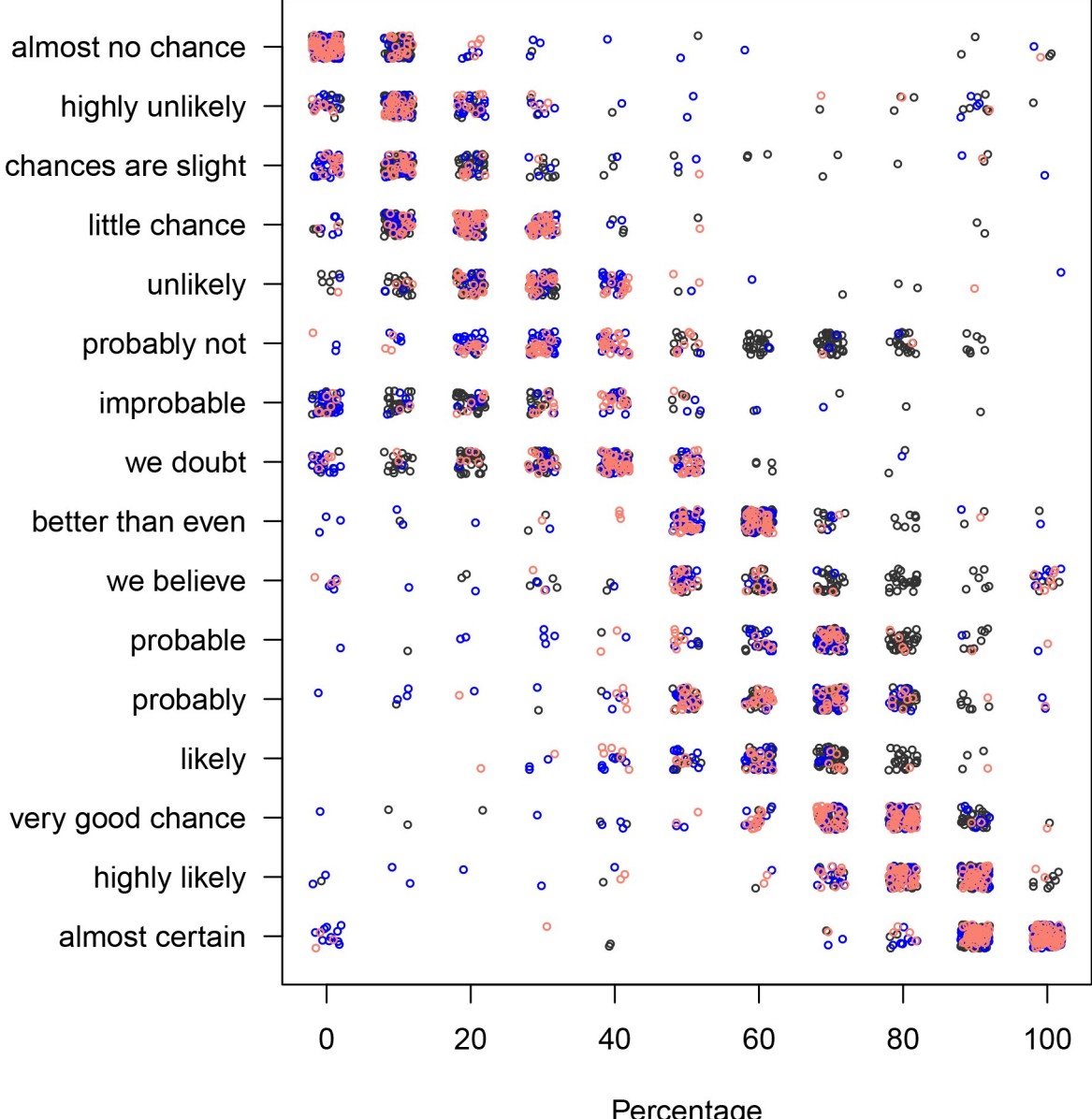

**Fig 5. The phrases associated with the percentages, using the "uncertainty" prompt.** The black, blue, and red colours correspond to the context-free, user and car contexts, respectively. Each dot represents one response.

effects. Rather, the importance scores are based on random permutations (and thus may sometimes be negative) and reflect improvement in predictive accuracy if the variable is added to the already selected ones. The higher the importance value, the more effect the variable has on the response [26]. While the effect of phrase and percentage is high for all models, the effect of the context is more differentiated. Thus, for the percentage-to-sentence ($P \rightarrow S$) interpretation, once the presence or absence of the context was accounted for, adding agent and valence had almost zero effect on accuracy. For the sentence-to-percentage ($S \rightarrow P$) interpretation, context had more predictive power, and adding agent still had a non-negligible effect on the prediction accuracy of the model. In all cases, valence had very little additive effect.

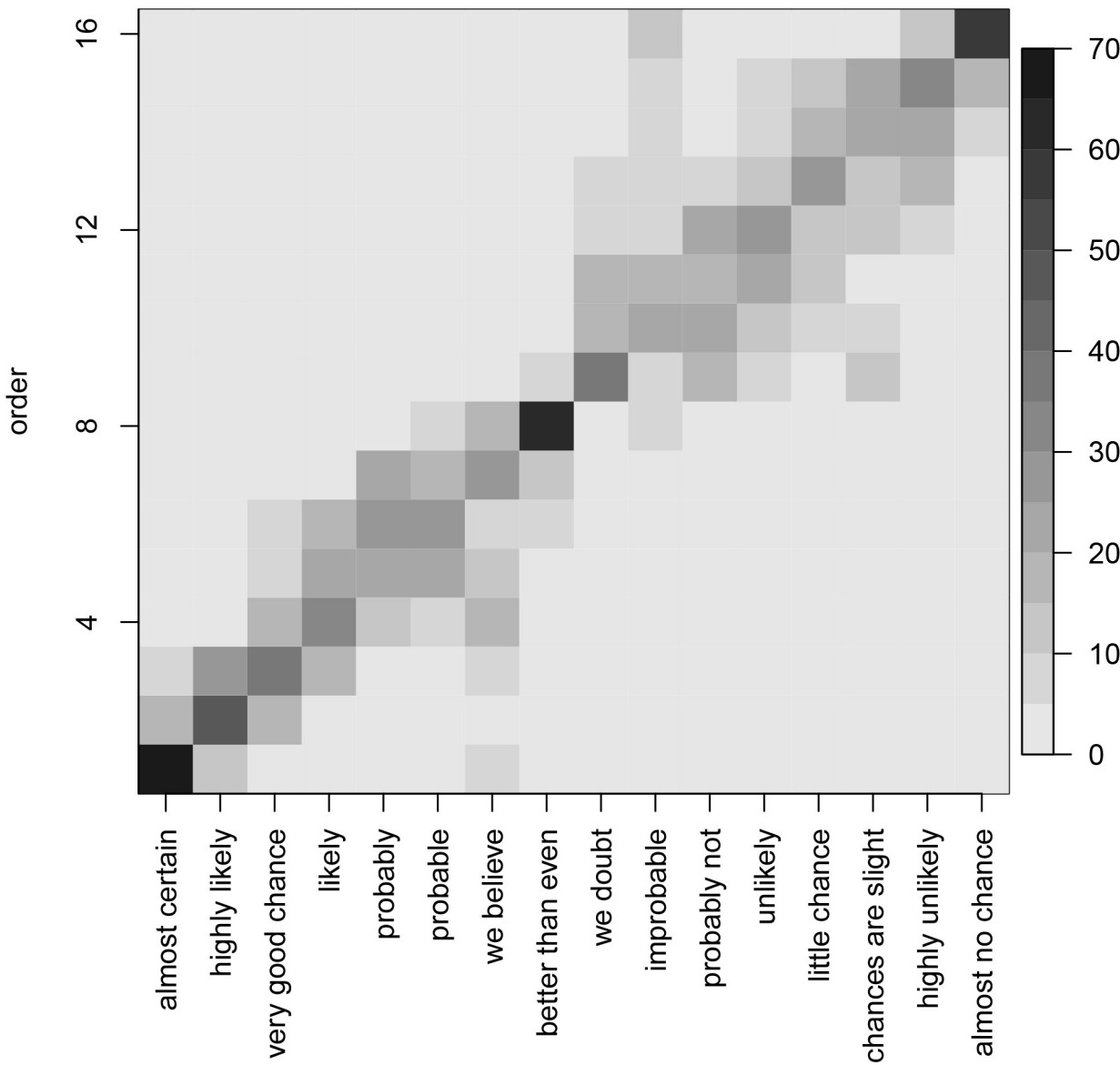

**Fig 6. The average order of the probability-related context-free phrases as perceived by respondents with the "likelihood" prompt.**

We have also run the permutation test to evaluate the p-values for these effects, and found them all to be significant at $p < .001$ level. This is to be expected for large data sets (see, for example, [38]. However, this probably just goes to show that statistical significance does not always translate into practical significance [39].

The decision rules for the percentage-to-sentence model are listed in Tables 2 and 3. The only time a variable other than percentage makes an appearance is in the Table 3, where, when the percentage is between 15% and 35%, the most likely response depends on whether the question was context-free. Other than that, only the percentage seems to matter.

The decision trees for predicting percentage from sentence are shown in Figs 9 and 10. At each node, a binary yes-or-no decision is made, and the corresponding branch is taken. Consider, for example, a 50-year-old woman, being asked to assign percentage to the context-free

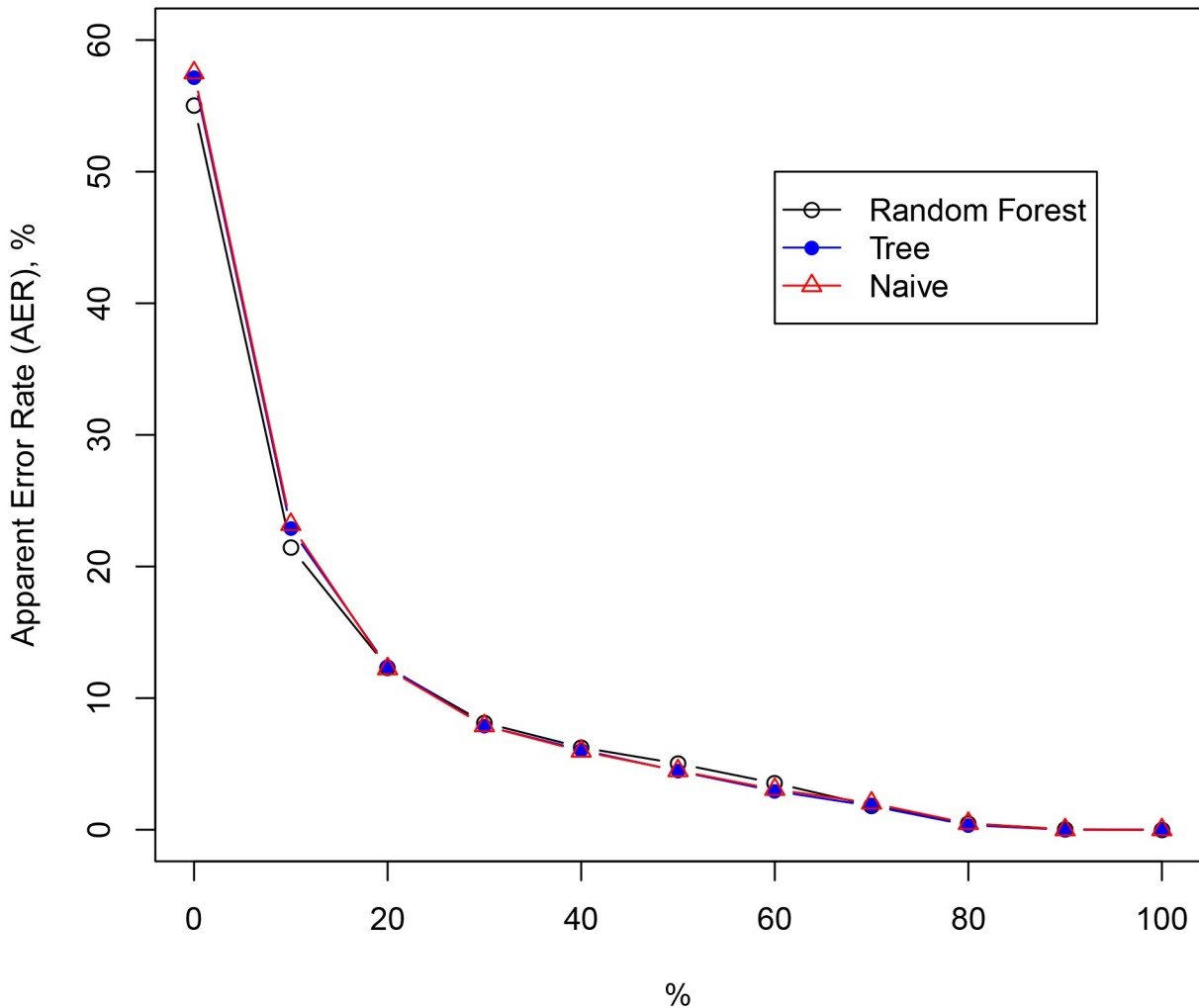

**Fig 7. The apparent error rate (AER) to within 1,2, etc. categories for the sentence-to-percentage responses with the likelihood prompt.**

phrase containing the words "almost no change" ($S = 16$). In this case, we would take the left branch at each split and ended up with 10%. The fact that age and agent variables are present in the trees show that they have an effect on prediction accuracy and thus on the respondents' perception of the probability phrases.

## Conclusions

We identified four decision trees, one for each combination of direction (percentage-to-sentence and sentence-to-percentage) and prompt ("uncertainty" and "likelihood"). These decision trees allow us to predict what selection the participants would make given certain information available to them. Variables that have predictive power, such as the phrase, are included in the tree and variables that have no predictive power are completely excluded from the tree, such as valence.

Our models show that given the sentence, we can predict the percentage response exactly around 50% of the time and within 10% of the true response 80% of the time (see Fig 7). Also, given the percentage, we can predict the exact phrase around 25% of the time and within the

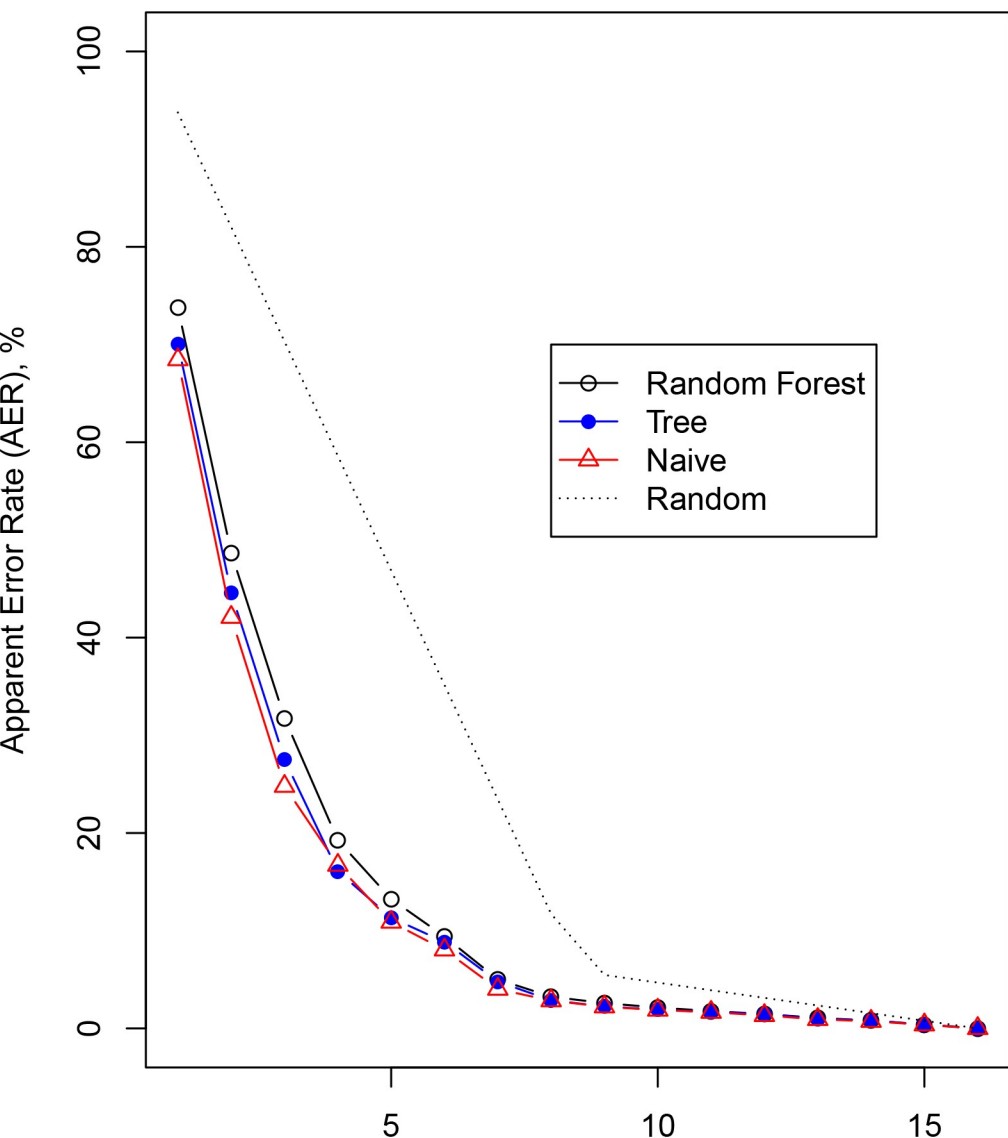

**Fig 8. The apparent error rate (AER) to within 1,2, etc. categories for the percentage-to-sentence responses with the likelihood prompt.** The categories are arranged as in Fig 6.

**Table 1. Importance scores of random forest variables evaluated as mean change in accuracy, conditional on the variables already included in the model (in order from top to bottom).** Higher values correspond to more important/influential variables. Here, *P* refers to percentage and *p* to the phrase.

| Variable | $P \rightarrow S$, cert | $P \rightarrow S$, lik | $S \rightarrow P$, cert | $S \rightarrow P$, lik |
|---|---|---|---|---|
| P (percentage) | 0.226 | 0.196 | | |
| p (phrase) | | | 0.283 | 0.223 |
| context | 0.010 | 0.010 | 0.015 | 0.013 |
| agent | 0.003 | 0.002 | 0.035 | 0.023 |
| valence | 0.001 | -0.002 | -0.004 | -0.001 |
| gender | -0.001 | -0.003 | 0.036 | 0.017 |
| age | -0.006 | -0.011 | 0.049 | 0.024 |

**Table 2. The decision rule for predicting phrase from percentage (P) with the "likelihood" prompt.** The phrases and their numbers are as listed in the Process section.

| Rule | Phrases |
|---|---|
| $P < 15$ | almost no chance |
| $15 < P < 25$ | little chance |
| $25 < P < 45$ | probably not |
| $45 < P < 65$ | better than even |
| $65 < P < 85$ | very good chance |
| $P > 85$ | almost certain |

adjacent category around 40% of the time. Note, that random guessing will succeed only 6% and 18%, respectively, see Fig 8. We can therefore conclude that the association between phrases and percentages is not random.

Moreover, there appears to be a preferred joint ordering of phrases and percentages (see Fig 6). The phrases "almost certainly", "almost no chance" and "better than even" were very strongly associated with the $1^{st}$, $16^{th}$, and $8^{th}$ ranks, respectively. Over 70% of the respondents ranked them in that way. Other phrases' order was less consistent. Visual inspection of Fig 6 shows a clear diagonal pattern which indicates the preferred ordering. It also shows a considerable scatter in the top right quadrant. While we find some evidence of non-random ordering of the phrases, the scatter plots reveal a considerable spread translated into relatively high apparent error rates (AER) of our models, and only the distinction between above and below even odds seems to be reliable.

Only the likelihood model for percentage-to-sentence did not utilise any contextual information (see Table 2), implying that context does not affect people's perception of likelihood. None of the models utilised valence, implying that it has no predictive power. Only the certainty model for sentence-to-percentage utilised agent information (see Fig 10), distinguishing between "you" and "autonomous vehicle". The other two models (uncertainty model for percentage-to-sentence and likelihood model for sentence-to-percentage) only distinguished between the presence and absence of context, but did not depend on the type of the agent.

The only exception is the phrase "probably not" in the uncertainty model for sentence-to-percentage (see Fig 10), which returns 30% if context is present and 70% if it is not. This particular phrase also had a considerable spread in the answers from the participants (see Fig 4). We speculate that the use of a negative expression ('not') might have confused the participants. We would recommend refraining from using negative or even double negative expressions.

**Table 3. The decision rule for predicting phrase from percentage (P) with the "uncertainty" prompt.** The phrases and their numbers are as listed in the Process section.

| Rule | | Phrase |
|---|---|---|
| $P < 5$ | | almost no chance |
| $5 < P < 15$ | | highly unlikely |
| $15 < P < 35$ | context-free | unlikely |
| | context | little chance |
| $35 < P < 45$ | | we doubt |
| $45 < P < 65$ | | better than even |
| $65 < P < 85$ | | very good chance |
| $P > 85$ | | almost certain |

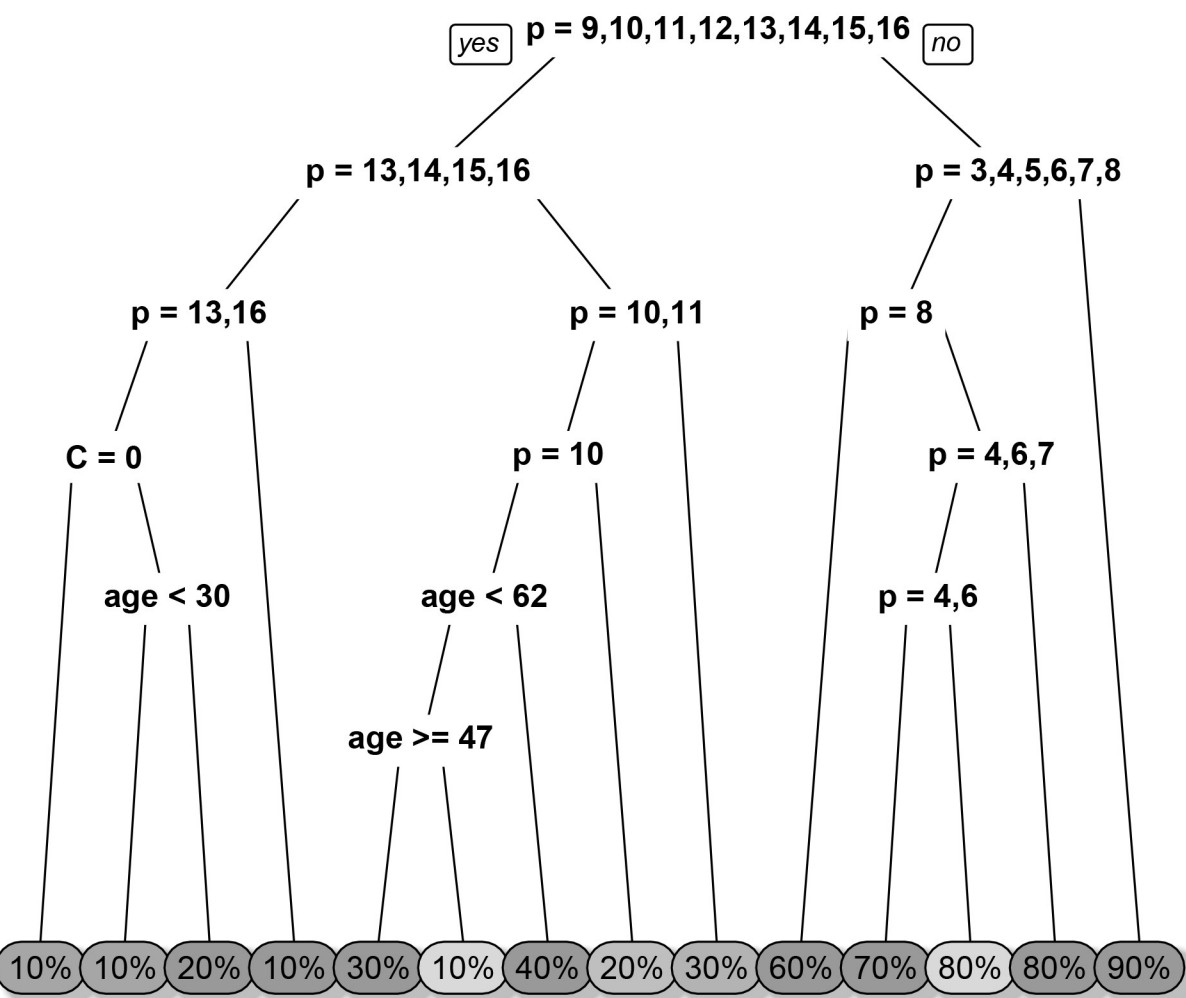

**Fig 9. The decision tree for sentence-to-percentage classification with the "likelihood" prompt.** *p* refers to the phrase ID, *C* refers to context (0 = no context, 1 = context), *A* refers to the agent (1 = you, 2 = autonomous vehicle) and *age* refers to the self-reported age of the participant. The left branches refer to "yes" while the right branches refer to "no" with respect to the logical conditions.

Although context and specific agent were included in the decision trees, they are only present in a small number of splits that result in adjacent categories. Their relative importance for the prediction results can thereby be described as moderate. Overall, we can conclude that context information, in particular valence, did have little influence on the predictive power of our models.

There are more points in the upper right quarter of Fig 4, which shows the likelihood model for sentence-to-percentage, compared with the certainty model for sentence-to-percentage as shown in Fig 2.

This might be due to the fact that some participants might have interpreted the word "certainty" as certainty about the probability of the event rather than equating the word "certainty" directly with the concept of probability. In this context, the phrases "almost certainly" and "almost no chance" would convey completely different probability (or, in laypeople's terms, perhaps, likelihood) of the event, but the same small level of uncertainty.

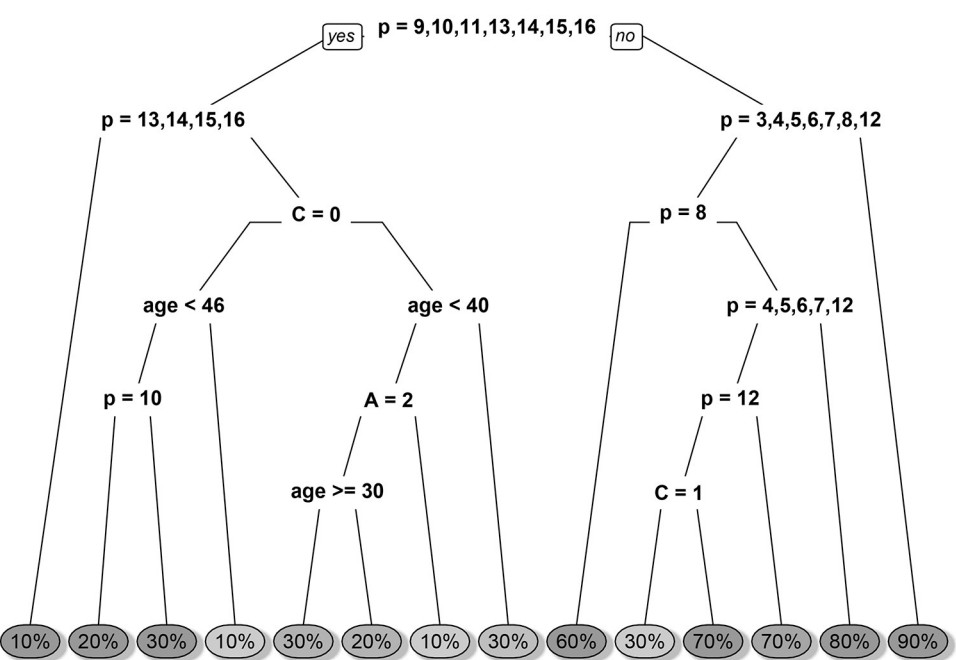

**Fig 10. The decision tree for sentence-to-percentage classification with the "uncertainty" prompt.** *p* refers to the phrase ID, *C* refers to context (0 = no context, 1 = context), *A* refers to the agent (1 = you, 2 = autonomous vehicle) and *age* refers to the self-reported age of the participant. The left branches refer to "yes" while the right branches refer to "no" with respect to the logical conditions.

From a strictly statistical perspective, given a binary yes/no event with probability of occurrence $p$, the uncertainty can be expressed via variance $p(1 − p)$ making the events with probability of occurrence $p = 0.50$ associate with the highest possible uncertainty. The responses of some of our participants, reflected in the upper right quarter of Fig 4 seem to support this statistical interpretation. It would be of interest to conduct further studies to better understand this observation.

The results of this study show that while it is in principle possible to communicate probability through the use of words, the consistency of the association of words to percentages remains far from perfect. It has to be pointed out that using numerical values is also not a perfect solution since people also struggle with understanding their meaning [9]. [40] went as far as to argue that different people associate very different meanings the phrases used in this study. He did, however, no offer an empirical study to justify his conclusion. [41] mentions the original study of [37] but also does not offer any empirical results.

What this study can provide is an empirical verified association of phrases to percentages and vice versa. When HRI researchers are asked to use words to help express probability then the results of this study will help them choose the most appropriate phrase. As the late Wittgenstein pointed out, the meaning of words is determined by their usage.

It would, however, seem wise to combine words with numerical values to offer the best possible chance of facilitating the correct understanding and interpretation of uncertainty. Negative and even double negative expression of uncertainty should be avoided at all times. We hope that this paper will enable HRI researchers to better communicate uncertainty in their interaction with participants, the media and policy makers.

## Limitations

There are several limitations to this study. First of all, we conducted this survey online. It could be argued that we might have received different results if we had asked participants the exact same questions while sitting in a Tesla that drives in autopilot mode. The participants might have given more importance to the valence of the statements.

It can even be argued that conducting the experiment in a controlled lab environment might have resulted in slightly different results since participants might have been more inclined to put more effort into answering the questions. The anonymity of the internet on which MTurk operates has the potential for participants offering only superficial responses. Participants might have considered the experiment as a puzzle that they need to solve and they believed that the valence information had no helpful information to solve the puzzle. While this is a well-known danger when using MTurk for recruiting participants, MTurk also offers some advantages, such as a more diverse sample.

Moreover, responding to a questionnaire and taking actions are not necessarily the same thing. Deciding to use an autonomous driving mode does entail risk and understanding the uncertainty around it is only the first step. What is more important is to see how this understanding changes the behaviour of the people. Once autonomous vehicles have a proven track record of safer than human driving we would hope that many would decide to use autopilots. This would lower the death toll on our roads.

We also have to point out that participants only answered a controlled subset of all possible questions. This was necessary since the total number of possible question combinations would have been 512, which would have been far too long. We would have had to deal with fatigue and the quality of the responses would have suffered.

## Future work

The 16 phrases taken from [37] might have not been the best possible choice. It might be interesting to offer the participants the option of providing their own phrases to express uncertainty. It is conceivable that the actively used vocabulary to express uncertainty is more constrained and hence more suitable for communication. Furthermore, certain words, such as marginal or significant, have specific meanings in the scientific community that would not be obvious to the general public.

## Author Contributions

**Conceptualization:** Christoph Bartneck, Elena Moltchanova.

**Data curation:** Christoph Bartneck, Elena Moltchanova.

**Formal analysis:** Elena Moltchanova.

**Investigation:** Christoph Bartneck.

**Methodology:** Christoph Bartneck, Elena Moltchanova.

**Project administration:** Christoph Bartneck.

**Resources:** Christoph Bartneck.

**Software:** Elena Moltchanova.

**Validation:** Elena Moltchanova.

**Visualization:** Elena Moltchanova.

**Writing – original draft:** Christoph Bartneck, Elena Moltchanova.

**Writing – review & editing:** Christoph Bartneck, Elena Moltchanova.

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
