## [Decision Letter · Decision Letter 0]

19 May 2020

PONE-D-20-01258

Expressing Uncertainty In Human-Robot Interaction

PLOS ONE

Dear Dr. Bartneck,

Thank you for submitting your manuscript to PLOS ONE. After careful consideration, we feel that it has merit but does not fully meet PLOS ONE’s publication criteria as it currently stands. Therefore, we invite you to submit a revised version of the manuscript that addresses the points raised during the review process.

The three reviewers have recognized the value and conceptual advance of your work. However, some specific points have to be dealt with before your manuscript can be accepted for publication.

We would appreciate receiving your revised manuscript by Jul 03 2020 11:59PM. To enhance the reproducibility of your results, we recommend that if applicable you deposit your laboratory protocols in protocols.io, where a protocol can be assigned its own identifier (DOI) such that it can be cited independently in the future. For instructions see: http://journals.plos.org/plosone/s/submission-guidelines#loc-laboratory-protocols

We look forward to receiving your revised manuscript.

Kind regards,

Roland Bouffanais, Ph.D.

Academic Editor

PLOS ONE

2. On line 190, please clarify if you meant to write "it is probable that you..." instead.

4. Thank you for stating the following in your Competing Interests section:  "None"

Reviewers' comments:

Reviewer's Responses to Questions

**Comments to the Author**

1. Is the manuscript technically sound, and do the data support the conclusions?

Reviewer #1: Partly

Reviewer #2: Yes

Reviewer #3: Yes

2. Has the statistical analysis been performed appropriately and rigorously? 

Reviewer #1: Yes

Reviewer #2: Yes

Reviewer #3: Yes

3. Have the authors made all data underlying the findings in their manuscript fully available?

Reviewer #1: Yes

Reviewer #2: Yes

Reviewer #3: No

4. Is the manuscript presented in an intelligible fashion and written in standard English?

Reviewer #1: Yes

Reviewer #2: Yes

Reviewer #3: Yes

5. Review Comments to the Author

Reviewer #1: In the manuscript, the authors proposed to associate words/phrases with percentage numbers to explore whether language might be a better way to communicate uncertainties compared to numbers. The idea is very interesting and in general the paper is well written. User studies on Amazon Mechanical Turk were conducted with good analysis.

One major concern from me is that the conclusion is very weak. Through the user studies, there seems no clear conclusion on which way is better in expressing uncertainties. The decision trees that the authors proposed did not outperform the naive method. Hence, I think that the authors should try to explore more aspects of the work and make it contribute more to the community.

Some minor comments focus on the writing in the conclusion part. Some of the expressions are very confusing to read. For instance, I cannot inteprete the exact meaning of "we can predict the percentage response exactly around 50% of the time and within 10% of the true response 80% of the time". The authors should summarize their results and findings in a more clear way.

Reviewer #2: The article is clear and well-written. The authors present a statistical understanding of the how people perceive percentages compared to a set of phrases. The authors do a great job explaining the reason for their methodology and the description of the experiment is easy to follow. The results are presented clearly and their significance is discussed. The conclusions follow from the results they present.

Make sure it is clear when presenting percentages that the meaning of the measurement. For example, a very brief explanation of what the apparent error rate is measuring could help understanding. Discuss how the decision tree is constructed in more detail. More discussion of the low p-values (line 304) would be useful. I would make it more clear why more emphasis is put on the other statistical measures instead of the p-value. You could also discuss more about the percentage values in Fig 9 and 10.

Minor comments: In line 190, there may be a typo in the prompt as it is not grammatically correct. I’d recommend including the abbreviation (AER) after “apparent error rate” at the beginning of sentence in line 284 for ease of reading. The sentence on line 362 which starts “There appears to be more points…” is a strange phrase to use since it could be quantitatively determined whether or not the upper quartile of Fig 4 has more points than Fig 2.

Reviewer #3: The underlying data is not available to the reviewer in the time of the review, e.g. through a provisional DOI or a private reviewer URL link.

Thus, point 3 on data availability is answered as No.

The authors in their submission state that "they will publish the underlying data with the Open Science Foundation (https://osf.io/) upon the acceptance of our manuscript."

The authors investigate how expressions of uncertainty such as "likely" and "improbable" are understood by non-experts or general public, so that uncertainty can be better communicated by Human-Robot Interaction researchers.

Thus, the authors evaluate how consistently expressions of uncertainty are interpreted as probability or likelihood values and the other way around, and whether other factors such as the context of the expression affect the interpretations.

For this purpose, the authors conduct a crowdsourced user study using the working context of autonomous vehicles.

Then, they employ non-parametric Classification and Regression Trees (CART) to model and reason about the considered factors in order to see if they affect the interpretations.

The results indicate (i) expressions of uncertainty are not perfectly interpreted as numbers and thus (ii) the expressions of uncertainty should be supplemented with numbers to enable correct interpretation (iii) negative expressions of uncertainty like "probably not" are less consistently associated and hence should be avoided.

The paper is well written and concise.

Please consider my following comments that focus on specific points.

Major comments:

The link on how does the study relate to previous work could be made clearer.

In particular, are there other work that follow on the reference 30, that which the phrases used to build uncertainty expressions are taken from?

Amazon Mechanical Turk (MTurk) is used to sample the general population.

In addition to seeing whether sex and age impact the interpretations, it could also be relevant to check their familiarity with probability theory, e.g. fundamentals like probabilities summing to one, or their background, profession or education.

Minor comments:

- lines 94-95, the example expression seems to present high certainty for the event that a vehicle "avoids" collision is unlikely to occur.

Is this the intended meaning, i.e. is it rather meant for the collision itself as an event being unlikely?

- Fig 1 could be replaced with the high resolution original, and also credit the original work of license CC-BY and the two authors.

at https://ourworldindata.org/uploads/2019/05/Causes-of-death-in-USA-vs.-media-coverage.png

- for naming consistency, it could be better to refer to the "uncertainty" prompt always as uncertainty (rather than in some places "certainty").

- for naming consistency, Fig. 9 and Fig. 10 could use the same indices while referring to the agents (0, 1 vs. 1, 2).

- line 56, typo: rendering "<"0.01%

- line 59, typo : Previous work~ed~

- line 65, could add AMT as an alternative abbreviation i.e. Amazon Mechanical Turn (AMT, MTurk)

- lines 73-74: the work is by Robert Sparrow and Mark Howard

- lines 47-48, should reference the Wittgenstein's quote from Tractatus.

- line 289 and caption of Fig. 8: "Figure" 6.

- line 364 - show"n"

- in the references , few punctuation-related issues e.g. ";." in 16 and 18, missing punctuation etc.

6. PLOS authors have the option to publish the peer review history of their article (what does this mean?). If published, this will include your full peer review and any attached files.

Reviewer #1: No

Reviewer #2: No

Reviewer #3: No

---

## [Author Response · Author response to Decision Letter 0]

26 May 2020

We included a dedicated response to the reviews letter in our submission. In there we address all the issues raised.

---

## [Decision Letter · Decision Letter 1]

15 Jun 2020

Expressing Uncertainty In Human-Robot Interaction

PONE-D-20-01258R1

Dear Dr. Bartneck,

We’re pleased to inform you that your manuscript has been judged scientifically suitable for publication and will be formally accepted for publication once it meets all outstanding technical requirements.

Kind regards,

Roland Bouffanais, Ph.D.

Academic Editor

PLOS ONE

Additional Editor Comments (optional):

Reviewers' comments:

Reviewer's Responses to Questions

**Comments to the Author**

1. If the authors have adequately addressed your comments raised in a previous round of review and you feel that this manuscript is now acceptable for publication, you may indicate that here to bypass the “Comments to the Author” section, enter your conflict of interest statement in the “Confidential to Editor” section, and submit your "Accept" recommendation.

Reviewer #2: All comments have been addressed

Reviewer #3: All comments have been addressed

2. Is the manuscript technically sound, and do the data support the conclusions?

Reviewer #2: Yes

Reviewer #3: Yes

3. Has the statistical analysis been performed appropriately and rigorously? 

Reviewer #2: Yes

Reviewer #3: Yes

4. Have the authors made all data underlying the findings in their manuscript fully available?

Reviewer #2: Yes

Reviewer #3: Yes

5. Is the manuscript presented in an intelligible fashion and written in standard English?

Reviewer #2: Yes

Reviewer #3: Yes

6. Review Comments to the Author

Reviewer #2: (No Response)

Reviewer #3: The reviewer has no more comments, and thanks to the authors.

Very minor: A typo at Line 132, "intersted".

7. PLOS authors have the option to publish the peer review history of their article (what does this mean?). If published, this will include your full peer review and any attached files.

Reviewer #2: No

Reviewer #3: No

---

## [Editor Report · Acceptance letter]

23 Jun 2020

PONE-D-20-01258R1 

Expressing Uncertainty In Human-Robot Interaction 

Dear Dr. Bartneck:

I'm pleased to inform you that your manuscript has been deemed suitable for publication in PLOS ONE. Congratulations! Your manuscript is now with our production department. 

Kind regards, 

on behalf of

Professor Roland Bouffanais 

Academic Editor

PLOS ONE